# Combined Biomarker System Predicts Prognosis in Patients with Metastatic Oral Squamous Cell Carcinoma

**DOI:** 10.3390/cancers15204924

**Published:** 2023-10-10

**Authors:** Tatjana Khromov, Lucas Fischer, Andreas Leha, Felix Bremmer, Andreas Fischer, Henning Schliephake, Michal Amit Rahat, Phillipp Brockmeyer

**Affiliations:** 1Department of Clinical Chemistry, University Medical Center Goettingen, 37075 Goettingen, Germany; tatjana.khromov@med.uni-goettingen.de (T.K.); andreas.fischer@med.uni-goettingen.de (A.F.); 2Department of Urology, University Medical Center Goettingen, 37075 Goettingen, Germany; lucas.fischer@med.uni-goettingen.de; 3Department of Medical Statistics, University Medical Center Goettingen, 37075 Goettingen, Germany; andreas.leha@med.uni-goettingen.de; 4Institute of Pathology, University Medical Center Goettingen, 37075 Goettingen, Germany; felix.bremmer@med.uni-goettingen.de; 5Department of Oral and Maxillofacial Surgery, University Medical Center Goettingen, 37075 Goettingen, Germany; schliephake.henning@med.uni-goettingen.de; 6Immunotherapy Laboratory, Carmel Medical Center, Haifa 3436212, Israel; mrahat@technion.ac.il; 7Ruth and Bruce Rappaport Faculty of Medicine, Technion-Israel Institute of Technology, Haifa 3109601, Israel

**Keywords:** oral squamous cell carcinoma, prognosis, overall survival, disease-free survival, connexin 43, EMMPRIN, E-cadherin, vimentin, metastasis, epithelial-to-mesenchymal transition

## Abstract

**Simple Summary:**

Metastasis poses a significant challenge in the advancement of oral squamous cell carcinoma, and it is closely associated with the epithelial–mesenchymal transition of tumor cells. In the current study, we investigated the expression profiles of connexin 43 and EMMPRIN, along with the known epithelial–mesenchymal transition markers E-cadherin and vimentin, using immunohistochemistry throughout the metastatic process. Additionally, we examined their prognostic impact both as a combined marker system and individually. The findings of this study reveal that a combined biomarker system can reliably predict overall and disease-free survival. Notably, alterations in EMMPRIN expression were found to have the highest prognostic impact, suggesting its potential as a therapeutic target for antimetastatic interventions.

**Abstract:**

Background: Metastatic oral squamous cell carcinoma (OSCC) is associated with poor patient prognosis. Metastasis is a complex process involving various proteins, tumor cell alterations, including changes attributable to the epithelial-to-mesenchymal transition (EMT) process, and interactions with the tumor microenvironment (TME). In this study, we investigate a combined protein marker system consisting of connexin 43 (Cx43), EMMPRIN (CD147), E-cadherin, and vimentin, with a focus on their roles in the invasive metastatic progression of OSCC and their potential utility in predicting prognosis. Methods: We conducted an immunohistochemical analysis to assess the protein expression profiles of Cx43, EMMPRIN, E-cadherin, and vimentin using tissue samples obtained from 24 OSCC patients. The metastatic process was mapped through different regions of interest (ROIs), including adjacent healthy oral mucosa (OM), center of primary OSCC, invasive front (IF), and local cervical lymph node metastases (LNM). The primary clinical endpoints were disease-free survival (DFS) and overall survival (OS). Results: Substantial changes in the expression profiles of the different marker proteins were observed among the different ROIs, with all *p*-values < 0.05, signifying statistical significance. Multivariable Cox regression analysis results showed a significant effect of increased EMMPRIN expression toward the IF on DFS (*p* = 0.019) and OS (*p* = 0.023). Furthermore, the combined predictive analysis showed a significant predictive value of the marker system for DFS (*p* = 0.0017) and OS (*p* = 0.00044). Conclusions: The combined marker system exhibited a significant ability to predict patient prognosis. An increase in EMMPRIN expression toward the IF showed the strongest effect and could be an interesting new antimetastatic therapy approach.

## 1. Introduction

Oral squamous cell carcinoma (OSCC) accounts for the majority of head and neck cancers and ranks among the most prevalent cancers worldwide [1]. Lymphatic metastasis, which is associated with poor patient prognosis, is common in advanced tumor stages [2].

Tumor cell dissemination requires the activation of the epithelial-to-mesenchymal transition (EMT) process. This promotes the loss of basal–apical polarity, breaks down tight and adhesive junctions, and allows the gain of motility in a mesenchymal spindle-like morphology. Physiologically, EMT occurs during development and wound healing [3]. However, in tumor cells, it is associated with invasiveness, metastasis, and enhanced tumor cell plasticity [4]. The EMT process provides tumor cells with a high degree of motility, enabling them to navigate the extracellular matrix (ECM), invade lymphatic or blood vessels, and spread to regional lymph nodes or distant tissues [5]. Some of the disseminated tumor cells (DTCs) colonize a distant organ, where they are likely to remain dormant [6]. 

Metastasis initiates when DTCs counteract quiescent signals in the local microenvironment, such as TGFβ2, BMP4, and BMP7 [7]. They then revert to the epithelial phenotype through the mesenchymal-to-epithelial transition (MET) program. This stimulates proliferation, leads to macrometastasis formation, and can result in cancer recurrence [5,8]. Throughout this process, metastatic cells exhibit plasticity, exhibiting varying phenotypes at different stages of this process.

Connexins (Cxs) are transmembrane proteins essential for gap junction (GJ) formation. They enable the direct passage of small molecules, such as ions, second messengers, metabolites, and microRNAs, mediating intercellular communication (GJIC) [9]. Additionally, they mediate the transport of cytosolic molecules to the extracellular milieu, playing a pivotal role in cellular homeostasis, as well as cell growth and development [9]. Among them, Cx43 is the most recognized human Cx protein [10], and our research indicates it serves as an independent prognostic factor in OSCC [11]. Cx43 expression can change during tumor progression. During EMT, tumor cells decrease their membrane-bound Cx43 expression, facilitating cell detachment and increasing cell motility [12]. However, during implantation and MET, they increase it to support tumor cell contacts via GJs [13], interacting with the endothelial barrier and surrounding cells in the TME [14,15]. Cxs and GJIC exhibit both pro- and antiproliferative effects, depending on the cell type and microenvironment. This is partly due to the exchange of molecules such as ATP, cAMP, or specific miRNAs between tumor cells and TME cells [16]. These functions may vary based on tumor type, tumor stage, interacting cell types, Cx molecule subtype, and their expression levels [17]. Cx43 within cells is also linked to microtubules, migration regulation, and apoptosis inhibition [18,19,20]. Overexpression of Cx43 in MDA-MB-231 breast cancer cells increases the expression of the epithelial markers E-cadherin and ZO-1, whereas Cx43 knockdown leads to the appearance of the mesenchymal protein N-cadherin [21], establishing a connection between Cx43 expression and MET. However, the exact role of Cx43 in EMT and MET has not been conclusively determined [22]. 

EMMPRIN/CD147 is a multifunctional transmembrane glycoprotein that mediates the interaction between tumor and stromal cells [23]. Overexpressed in over 70% of human tumors, its expression is associated with higher tumor grade and stage, metastasis, and poor prognosis [24]. EMMPRIN is primarily recognized for its proangiogenic role, triggering VEGF and MMPs through homophilic interactions. We have identified an epitope in its extracellular domain I that is responsible for both activities [25]. Additionally, EMMPRIN acts as a chaperone for lactate transporters MCT-1 and MCT-4, facilitating lactate efflux, which is vital for tumor cells primarily dependent on glycolysis. Increased extracellular lactate levels have been shown to upregulate mesenchymal markers such as vimentin and N-cadherin [26], suggesting an indirect association of EMMPRIN with EMT. Furthermore, EMMPRIN regulates hyaluronan synthesis and can bind to its receptor CD44, a recognized cellular stem cell biomarker that contributes to tumor cell invasiveness and chemoresistance [27]. In summary, these properties suggest that EMMPRIN plays roles in tumor cell metabolism, survival, proliferation, invasiveness, metastasis, and angiogenesis, and likely promotes EMT [28]. 

In this study, we mapped the metastatic process from healthy oral mucosa (OM) to solid lymph node metastasis (LNM) in OSCC. Using immunohistochemistry, we examined the expression profiles of Cx43 and EMMPRIN alongside the known EMT markers E-cadherin and vimentin and assessed the prognostic significance of all marker proteins and the combined marker system.

## 2. Materials and Methods

### 2.1. Patients

The sample size was determined using StatMate software (version 2, GraphPad Software, Boston, MA, USA). With a sample size of 24, we achieved 95% power to detect a difference between means of 56.31 with a significance level (alpha) of 0.05 (two-tailed). We utilized tumor tissue samples from 24 OSCC patients, primarily treated surgically between 2016 and 2019, for immunohistochemical evaluation. For assessment of baseline clinical characteristics, tumor stages T1 and T2 were grouped together, as were stages T3 and T4. In addition, American Joint Commission on Cancer (AJCC) clinical stages I and II were combined, as were stages III and IV. Regarding nodal status analysis, patients were categorized into lymph node positive and lymph node negative groups. The primary clinical endpoints were disease-free survival (DFS) and overall survival (OS). Before enrollment, patients provided written informed consent. The study adhered to the principles of the Declaration of Helsinki and received approval from a clinical ethics committee (approval no. 07/06/09, updated April 2018).

### 2.2. Tissue Sample Processing and Semiautomated Semiquantitative Immunohistochemical Analysis

Tumor tissue samples from patients were promptly obtained post-surgical resection, preserved in neutral buffered 4% formalin, and then embedded in paraffin. Immunohistochemical staining was performed on 2 µm sections using a fully automated slide stainer (Agilent Technologies, Santa Clara, CA, USA), as specified in Table 1.

Tissue slides were digitized at 20× magnification with a resolution of 0.5 μm/pixel using a Motic EasyScan One slide scanner (Motic, Hong Kong, China). For semiautomated semiquantitative immunohistochemical assessment, we employed the open-source image analysis software quPath [29]. To comprehensively map the metastatic process (Figure 1), we analyzed tissue samples from primary tumors (along with adherent healthy oral mucosa) and corresponding local cervical lymph node metastases (LNM). Different regions of interest (ROIs) were digitally defined, as illustrated in Figure 2, Figure 3, Figure 4 and Figure 5. For OSCC tissue samples, we defined three ROIs in the adjacent healthy mucosa, three ROIs in the center of the primary OSCC, and another three at the invasive front (IF). In LNM, three ROIs were distributed over the entire metastasis. Each ROI was defined to be approximately 1 cm^2^ in size.

The quPath cell detection algorithm was performed within all ROIs. The software was calibrated to differentiate between tumor and stromal cells. To enhance accuracy, this training was performed three times utilizing an artificial intelligence (AI) function. Using the software’s default settings, the immunohistochemically labeled marker proteins (Cx43, EMMPRIN, E-cadherin, and vimentin) were semiautomatically scored based on the percentage of tumor cells showing positive staining and signal intensity. The histoscore (H-score) was calculated by adding 3× the percentage of tumor cells with strong staining, 2× the percentage with moderate staining, and 1× the percentage of weak staining. This method yielded scores that ranged from 0 (all tumor cells negative) to 300 (all tumor cells strongly positive).

### 2.3. Statistical Analysis

To gauge the dynamics of protein expression across various ROIs, expression differences (Δ) were computed. For each protein, all pairwise expression differences between ROIs were determined. These differences were then assessed using pairwise contrast tests grounded on linear mixed-effects models, which modeled expression by ROI. The resulting estimates are reported with 95% confidence intervals and *p*-values adjusted using Tukey’s test. A full linear mixed-effect model for protein abundance was fitted with protein, tissue, and their interactions as predictors. Drawing from this model, we estimated the expected marginal means for the differences between ‘neighboring’ ROIs. To contrast each of these differences among all protein pairs, contrast tests were employed. The results, accompanied by Holm adjusted *p*-values, are presented alongside the expected marginal effects with their 95% confidence intervals. To assess potential associations between both clinical and protein expression data with DFS and OS, univariable Cox proportional hazards regression models were employed. Each model’s fit was evaluated against the null model using likelihood ratio tests. All protein variables with a *p*-value smaller than 0.1 from these likelihood ratio tests were rescreened for association with DFS and OS adjusted for AJCC stage and age. The limited number of patients and events limits the complexity of the models that can be fitted to the data. Given that the variables adjuvant therapy, pT, pN, and AJCC stage were highly correlated, the AJCC stage was selected to represent this group of variables. The output model coefficients were reported as hazard ratios (HR) accompanied by a 95% confidence interval and their corresponding *p*-value. For both prognostic endpoints (DFS and OS), a multivariable Cox regression model was fitted using only protein expression data, according to the following equations:

Score equation for DFS:0.73805 − 0.00759 * ΔEMMPRIN (OM-IF) + 0.00576 * ΔE-Cadherin (OM-IF) + 0.00783 * Cx43 (IF)

Score equation for OS:0.21709 + 0.01490 * ΔEMMPRIN (OCSS-IF) − 0.01966 * ΔVim (OM-OSCC) + 0.00531 * ΔCx43 (OM-IF)

For every protein, the ROI/ROI difference showing the most significant univariate association with survival (as determined by the *p*-value from the likelihood ratio tests) was selected. Given the limited sample size, only the top three markers were selected. The resulting model coefficients were reported as hazard ratios (HR) accompanied by a 95% confidence interval and their respective *p*-value. For visualization of the effect, model predictions were binarized at the median and maximally selected rank statistic. Subsequently, Kaplan–Meier curves were plotted in the resulting subgroups and compared using log-rank tests. All statistical tests were conducted with a significance level set at α = 5%. All analyses were performed with the statistical software R (version 4.1.2; R Core Team 2021) [30] using the R-package lme4 (version 1.1.28) [31] for the mixed effect regression models and emmeans (version 1.7.2) [32] for computing the expected marginal effects and performing contrast tests. The full dataset is available online at https://doi.org/10.25625/FNR4EX (accessed on 27 September 2023).

## 3. Results

### 3.1. Patients’ Clinical Baseline Characteristics

The patient population consisted of 14 male and 10 female OSCC patients, aged between 55 and 81 years. OSCC was diagnosed in various regions of the oral cavity, including the floor of the mouth, buccal mucosa, gingiva, inside of the lips, palate, and tongue. Out of the participants, 11 patients had an AJCC stage of II or lower, while 13 patients had an AJCC stage greater than II. Median overall survival was 3.5 years and median disease-free survival was 10.5 months. Table 2 details the baseline clinical attributes of all the patients.

### 3.2. Marker Protein Expression in Different Regions of Interest (ROIs)

Semiquantitative immunohistochemical analysis, depicted in Figure 2, Figure 3, Figure 4 and Figure 5 and summarized in Figure 1, demonstrated differential expression profiles for the different marker proteins. Notably, Cx43 and E-cadherin displayed similar patterns of expression, and likewise, EMMPRIN and vimentin shared comparable expression patterns.

**Figure 1 cancers-15-04924-f001:**
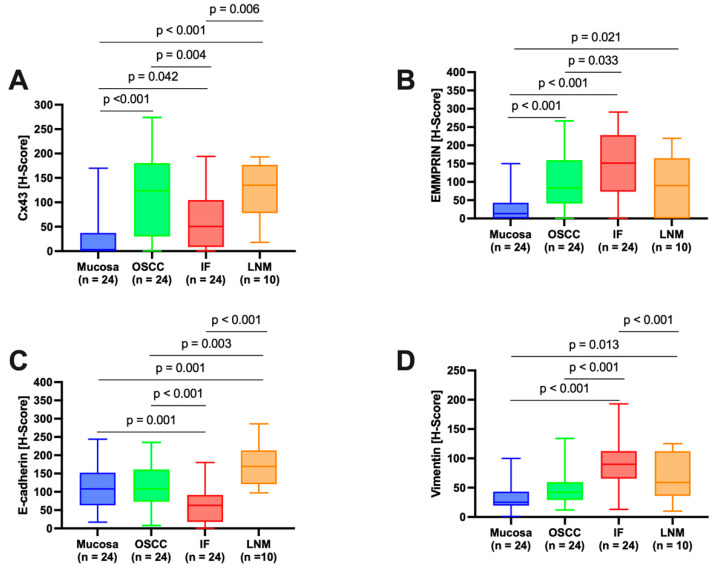
Quantification of marker protein expression across different regions of interest (ROIs) (oral mucosa; center of primary OSCC; invasive front, IF; and local lymph node metastasis, LNM) annotated with the pairwise comparison results. Histoscore (H-score) values are reported. The score is derived by adding 3× the percentage of tumor cells with strong staining, 2× those with moderate staining, and 1× those with weak staining. This generates a score range from 0 (all tumor cells negative) to 300 (all tumor cells strongly positive). (**A**) Cx43 expression; (**B**) EMMPRIN expression; (**C**) E-cadherin expression; (**D**) vimentin expression.

#### 3.2.1. Connexin 43 (Cx43)

Histological evaluation within each ROI revealed low Cx43 expression in the healthy OM (Figure 2A, ROI 1–3; Figure 2B). There was an increase in Cx43 in the center of the primary OSCC (Figure 2A, ROI 4–6; Figure 2C) and a decrease in the IF (Figure 2A, ROI 7–9; Figure 2D). In the corresponding LNM, there was a noticeable increase in Cx43 expression, predominantly in the outer regions of tumor cell growth (Figure 2E). 

Pairwise contrast tests of H-score values verified a significant increase in Cx43 expression from the OM to the center of the primary OSCC (*p* < 0.001). Conversely, there was a significant decrease in expression transitioning from the center of the primary OSCC to the IF (*p* = 0.004). A significant increase in Cx43 expression was observed in the LNM compared to the IF (*p* = 0.006, Figure 1A). Cx43 expression in the LNM was comparable to that observed in the center of the primary OSCC. The H-score values of the semiquantitative immunohistochemical evaluation of Cx43 expression in all ROIs are shown in Table 3.

**Figure 2 cancers-15-04924-f002:**
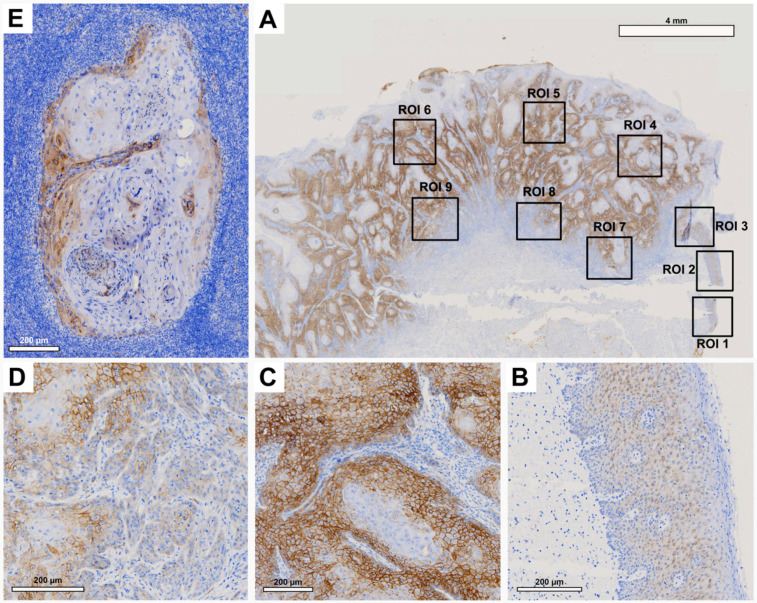
Representative illustration of immunohistochemical evaluation in primary OSCC tissue sample and its corresponding lymph node metastasis (LNM) based on Cx43 staining. (**A**) Overview of primary OSCC with individual ROIs. The upper right corner illustrates the transition to adjacent healthy oral mucosa (ROI 1–3). (**B**) Enlarged view of ROI 2 (OM), (**C**) ROI 6 = primary OSCC, and (**D**) ROI 9 = invasive front (IF). (**E**) Enlarged view of LNM. Scale bar, 4 mm and 200 μm.

#### 3.2.2. EMMPRIN

Histological evaluation revealed strong EMMPRIN expression in patient tissue samples (Figure 3). In healthy OM, EMMPRIN was most abundant in the lower epithelial layers approaching the basement membrane (Figure 3B). There was a noticeable increase in EMMPRIN expression at the center of the primary OSCC (Figure 3C), which was particularly pronounced toward the IF (Figure 3D). In contrast, lower EMMPRIN expression was observed in the corresponding LNM (Figure 3E).

Pairwise contrast tests of EMMPRIN expression within the different ROIs revealed a significant increase in the expression profile from the healthy OM to the center of the primary OSCC (*p* < 0.001), with a further significant increase toward the IF (*p* = 0.033). In contrast, while EMMPRIN expression in the LNM decreased again, the decrease was not statistically significant in comparison to the IF (*p* = 0.122), aligning to levels observed in the center of the primary OSCC. Table 4 presents the H-score values from the semiquantitative immunohistochemical evaluation of EMMPRIN expression across all ROIs.

**Figure 3 cancers-15-04924-f003:**
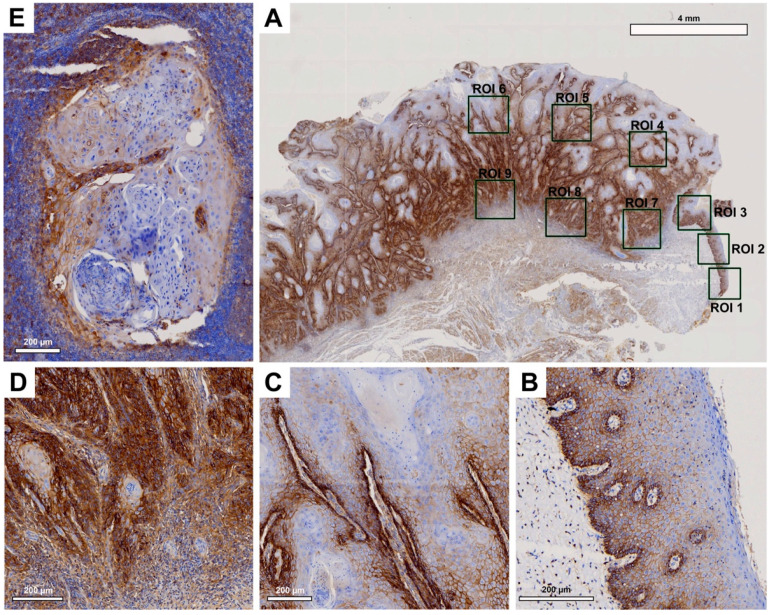
Representative illustration of immunohistochemical evaluation in primary OSCC tissue sample and corresponding lymph node metastasis (LNM) based on EMMPRIN staining. (**A**) Overview of primary OSCC with individual ROIs. The upper right corner illustrates the transition to adjacent healthy oral mucosa (ROI 1–3). (**B**) Enlarged view of ROI 2 (OM), (**C**) ROI 6 = primary OSCC, and (**D**) ROI 9 = invasive front (IF). (**E**) Enlarged view of LNM. Scale bar, 4 mm and 200 μm.

#### 3.2.3. E-Cadherin

From a histological standpoint, patient tissue samples exhibited comparable membranous E-cadherin expression both in the OM and at the center of the primary OSCC (Figure 4B,C). Notably, there was a significant decline in expression moving toward the IF (Figure 4D). In the corresponding LNM, E-cadherin expression was again increased (Figure 4E), reaching levels higher than the OM or the center of the primary OSCC.

No significant difference was observed between E-cadherin expression in the healthy OM and the center of the primary OSCC (*p* = 0.980). However, a significant decrease in E-cadherin expression was observed between the center of the primary OSCC and the IF (*p* < 0.001). Moreover, a significant increase in E-cadherin expression was observed in LNM compared to the IF (*p* < 0.001) and the center of OSCC (*p* = 0.003). The H-score values of the semiquantitative immunohistochemical evaluation of E-cadherin expression in all ROIs are shown in Table 5.

**Figure 4 cancers-15-04924-f004:**
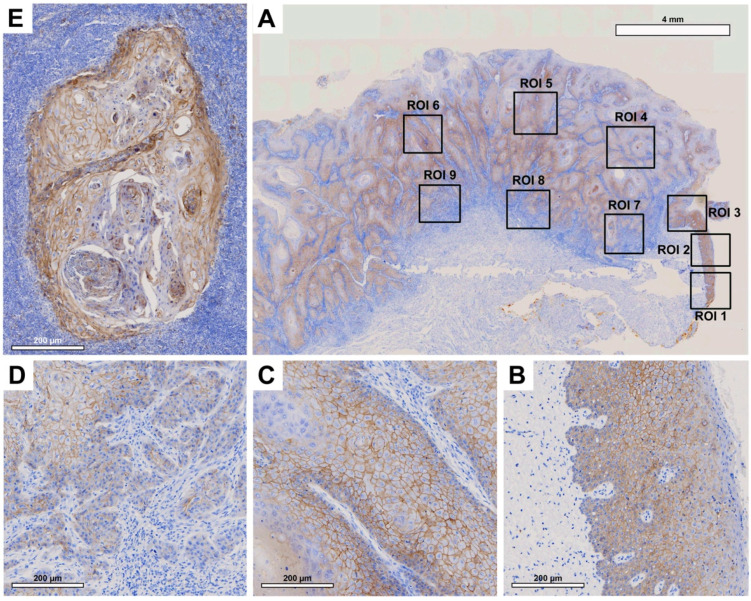
Representative illustration of immunohistochemical evaluation in primary OSCC tissue sample and corresponding lymph node metastasis (LNM) based on E-cadherin staining. (**A**) Overview of primary OSCC with individual ROIs. The upper right corner shows the transition to adjacent healthy oral mucosa (ROI 1–3). (**B**) Enlarged view of ROI 2 (OM), (**C**) ROI 6 = primary OSCC, and (**D**) ROI 9 = invasive front (IF). (**E**) Enlarged view of LNM. Scale bar, 4 mm and 200 μm.

#### 3.2.4. Vimentin

Histological evaluation revealed low vimentin expression in the healthy OM (Figure 5B). There was a slight increase at the center of the primary OSCC (Figure 5C), but a strong increase was observed toward the IF (Figure 5D). In the corresponding LNM, there was a noticeable decrease in vimentin expression. However, it was primarily localized at the junction with the surrounding tissue (Figure 5E), which was still higher than in the center of the primary OSCC.

The analysis of H-score values showed no significant difference between healthy OM and the center of the primary OSCC (*p* = 0.222). However, a significant increase in vimentin expression was observed between the center of the primary OSCC and the IF (*p* < 0.001). In LNM, vimentin expression was decreased compared to the IF; however, this difference did not reach statistical significance (*p* = 0.061). Table 6 presents the H-score values from the semiquantitative immunohistochemical assessment of vimentin expression across all ROIs.

**Figure 5 cancers-15-04924-f005:**
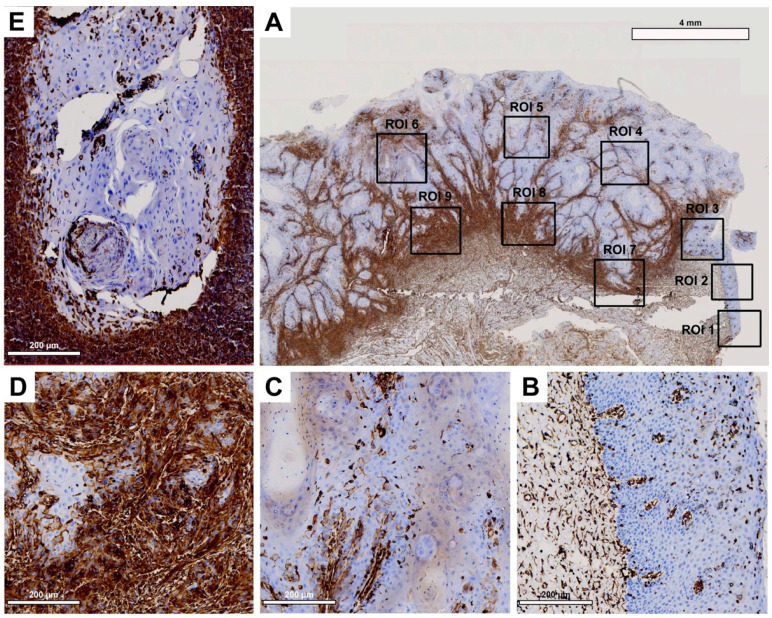
Representative illustration of immunohistochemical evaluation in primary OSCC tissue sample and corresponding lymph node metastasis (LNM) based on vimentin staining. (**A**) Overview of primary OSCC with individual ROIs. The upper right corner illustrates the transition to adjacent healthy oral mucosa (ROI 1–3). (**B**) Enlarged view of ROI 2 (OM), (**C**) ROI 6 = primary OSCC, and (**D**) ROI 9 = invasive front (IF). (**E**) Enlarged view of LNM. Scale bar, 4 mm and 200 μm.

### 3.3. Analysis of Independent Marker Protein Expression

Upon examining the independent protein expression profiles of the four marker proteins, no significant differences were observed between the expression of EMMPRIN and vimentin (all *p*-values > 0.05) across all ROIs. However, significant differences in the expression of Cx43 and E-cadherin were identified between the healthy oral mucosa and the center of the primary OSCC (*p* = 0.005), as well as between the oral mucosa and the IF (*p* = 0.002; Figure 6).

**Figure 6 cancers-15-04924-f006:**
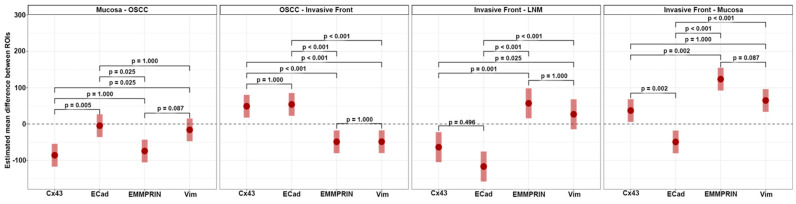
Differences between ‘neighboring’ ROIs (OM vs. OSCC, OSCC vs. IF, IF vs. LNM, IF vs. OM) were estimated and compared between proteins. This plot illustrates the expected marginal means of these differences (red dots) with 95% confidence intervals (red bars). The provided *p*-values originate from contrast tests adjusted for multiple comparisons using Holm’s procedure.

### 3.4. Patient Prognosis Prediction

#### 3.4.1. Disease-Free Survival (DFS)

The initial univariable screening analysis revealed that DFS was dependent on advanced age at initial diagnosis (>45 years), adjuvant therapy approach performed (radiation and/or chemotherapy), E-cadherin expression change from OM to IF, Cx43 expression within the IF, and both high EMMPRIN expression within the IF and its expression change from OM to IF. Table 7 presents all the significant factors identified in the univariable screening analysis.

All marker protein variables that yielded a *p*-value < 0.1 in the initial screening analysis were retested for their association with DFS. This reassessment was adjusted for the clinical parameters of AJCC stage and age at diagnosis. This analysis revealed a significant independent influence on DFS of high EMMPRIN expression in the center of the primary OSCC and in the IF, as well as EMMPRIN expression change from OM to IF. Additionally, a significant independent effect was observed for E-cadherin expression change from OM to IF and Cx43 expression within the IF (Table 8).

In the final multivariable Cox regression model, only the protein expression differences between ROIs with the best univariable association with DFS were selected. This analysis revealed a statistically significant effect of EMMPRIN expression change from OM to IF on DFS, while no statistically significant associations were observed for E-cadherin and Cx43 (Table 9).

The final multivariable Cox regression model was used for combined DFS prediction. Model predictions were binarized at the median, and Kaplan–Meier curves were plotted in the resulting subgroups and compared with log-rank tests. Significant bifurcation of the survival curves was observed (Figure 7).

**Figure 7 cancers-15-04924-f007:**
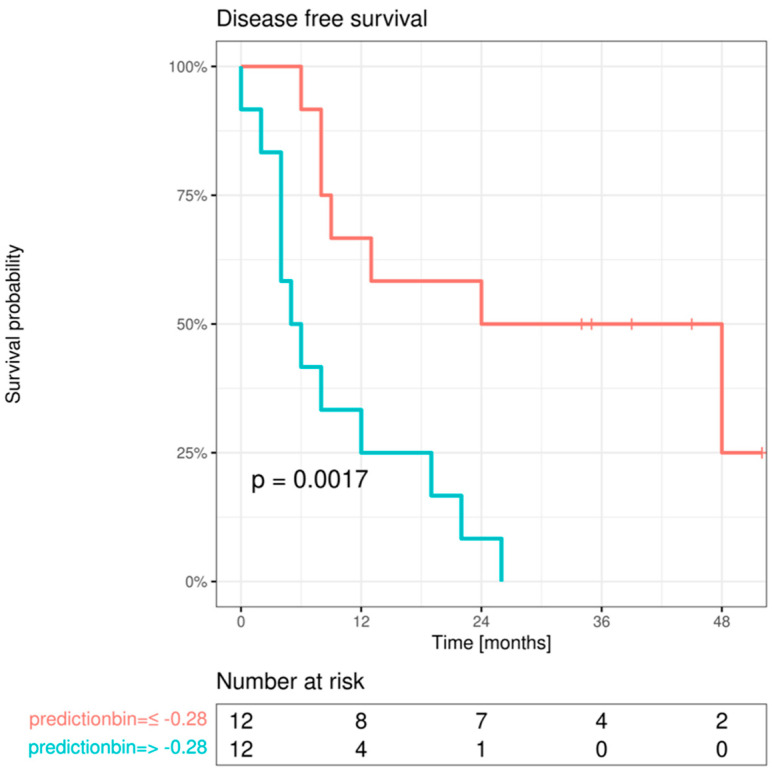
Survival curves for DFS after prediction by multivariable Cox regression model when split at the optimal cutoff. *p*-value from log-rank test.

#### 3.4.2. Overall Survival (OS)

Univariable screening was also performed for OS. Significant influences were found for high pT status (3/4), positive lymph node status (pN+), adjuvant therapy approach performed, high AJCC stage (III/IV), an increase in E-cadherin expression from IF to LNM, a decrease in E-cadherin expression from OM to IF, high vimentin expression in OM and in the center of the primary OSCC, an increase in vimentin expression from OM to the center of the primary OSCC, an increase in vimentin expression from OM to IF, high Cx43 expression in the IF, a decrease in Cx43 expression from OM to IF, high EMMPRIN expression in OM and IF, and an increase in EMMPRIN expression from the center of the primary OSCC to the IF. All statistically significant factors of the univariable screening analysis are shown in Table 10.

All marker protein variables that yielded a *p*-value < 0.1 in the initial screening analysis were retested for association with OS, adjusting for the clinical parameters of AJCC stage and age. This analysis revealed a significant independent influence on OS of EMMPRIN expression change from the center of the primary OSCC to the IF, vimentin expression change from OM to the center of the primary OSCC, and Cx43 expression change from OM to the IF (Table 11).

Multivariable Cox regression analysis revealed a significant effect of increasing EMMPRIN expression from the center of the primary OSCC to the IF on OS (Table 12).

To visualize the impact of EMMPRIN expression on patient OS, model predictions were binarized at the maximum selected rank statistics, and Kaplan–Meier curves were plotted in the resulting subgroups and compared with log-rank tests (Figure 8).

**Figure 8 cancers-15-04924-f008:**
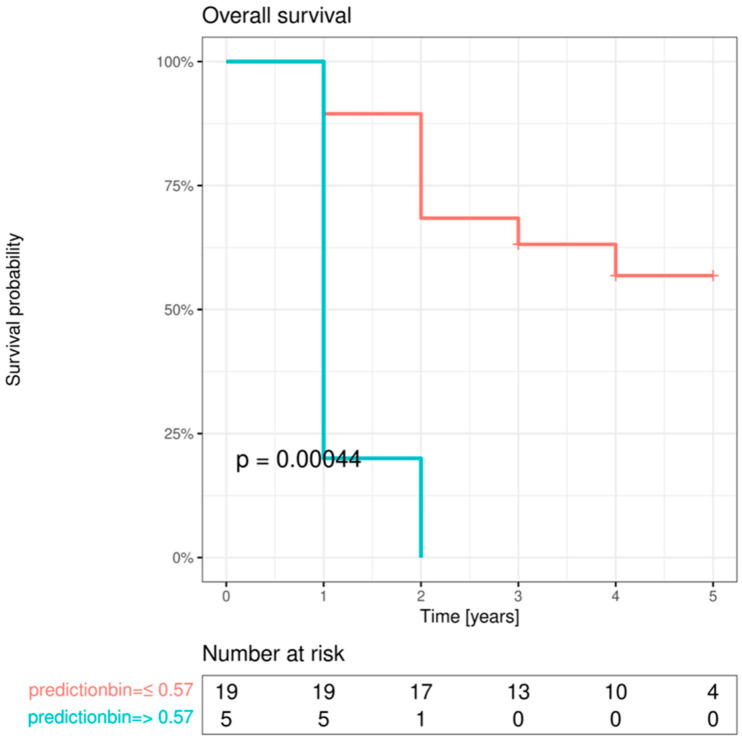
Survival curves for OS after prediction by multivariable Cox regression model when split at the optimal cutoff. *p*-value from log-rank test.

## 4. Discussion

Metastasis represents a milestone in OSCC progression [33], and it is associated with poor patient prognosis and decreased health-related quality of life [2]. In this study, we mapped the metastatic process from healthy oral mucosa to cervical lymph node metastasis in tissue samples from 24 OSCC patients. We evaluated the expression profiles of both transmembrane proteins Cx43 and EMMPRIN, along with the known EMT markers E-cadherin and vimentin, using immunohistochemistry. Moreover, we investigated the prognostic impact of each protein and tested whether a combined biomarker system could significantly predict the clinical endpoints DFS and OS.

Cx43 expression was detected in all analyzed tissue types, including oral mucosa, center of primary OSCC, invasive front, and lymph node metastases. Expression was at its lowest in oral mucosa and exhibited a significant increase as it progressed toward the center of the primary OSCC. Cx43 expression showed a decline toward the invasive front, followed by a subsequent increase in lymph node metastases. The expression profile of Cx43 paralleled that of the established epithelial marker E-cadherin [34]. Both Cx43 and E-cadherin are transmembrane proteins that are important for the formation of intercellular junctions, gap junction channels, and the maintenance of intercellular communication [15]. This may explain the increase in Cx43 toward the center of the primary OSCC, where cells proliferate and interact with each other. Toward the invasive front, where tumor cells undergo EMT to gain the ability to migrate and metastasize [35], intercellular junctions must be disrupted and Cx43 and E-cadherin expression are reduced [11]. In lymph node metastasis, where tumor cells communicate with each other again, both protein expression profiles (Cx43 and E-cadherin) are restored. This underscores the connection between Cx43 expression and mesenchymal-to-epithelial transition (MET) features. While our prior studies showed the independent prognostic significance of Cx43 in OSCC [11], we were unable to validate this effect in the present cohort using multivariable Cox regression analysis, possibly attributed to the limited number of cases in this study.

In this study, we demonstrated that EMMPRIN is highly expressed in both physiological oral mucosa and primary OSCCs. Similar results have been reported previously by Rajshri et al. [36] and Min et al. [37]. However, we demonstrated for the first time that EMMPRIN expression significantly increases toward the invasive front of OSCC and decreases in lymph node metastasis. The EMMPRIN expression profile corresponds to that of the well-known mesenchymal marker vimentin [38]. These findings imply that high EMMPRIN expression in migrating OSCC tumor cells plays a crucial role in facilitating invasion into the surrounding tissue and metastasis. This feature is not required for lymph node metastases, where the focus is on cellular cluster growth, and consequently, EMMPRIN expression is reduced, thereby associating EMMPRIN with EMT features. Furthermore, among all the markers examined, EMMPRIN expression had the strongest prognostic impact on clinical endpoints, inducing DFS and OS, as measured through multivariable Cox regression analysis. We have recently shown that EMMPRIN plays a role in preventing tumor cells from entering the dormant state, and that knocking down EMMPRIN pushes cells toward dormancy [39]. Therefore, considering the potential of targeting EMMPRIN expression in metastatic OSCC tumor cells, this approach may hold promise as a novel antimetastatic therapy.

Furthermore, we provided evidence that the combined biomarker system of Cx43, EMMPRIN, E-cadherin, and vimentin possesses a significant predictive capacity for both DFS and OS in patients with metastatic OSCC. We suggest this combined system warrants comprehensive evaluation in a larger number of patients before considering its potential implementation in routine clinical practice.

## 5. Conclusions

Metastasis in oral squamous cell carcinoma is associated with poor patient prognosis. Altered EMMPRIN expression and localization toward the invasive front shows the highest influence on both disease-free survival and overall survival. This finding suggests that targeting EMMPRIN could present a promising avenue for new antimetastatic therapy approaches. A combined biomarker system consisting of Cx43, EMMPRIN, E-cadherin, and vimentin demonstrates the ability to reliably predict both disease-free survival and overall survival, potentially facilitating prognostic assessment within routine clinical practice.

## Figures and Tables

**Table 1 cancers-15-04924-t001:** Staining protocol.

Antigen	Antibody	Pretreatment	Detection Method	Source
E-cadherin	Mouse, monoclonal, clone NCH-38, RTU	HIER (pH 9)	Dako EnVision FLEX	Agilent Dako (IR05961-2)
Vimentin	Mouse, monoclonal, clone V9, RTU	HIER (pH 9)	Dako EnVision FLEX	Agilent Dako (IR63061-2)
Cx43	Rabbit, monoclonal, clone EPR21153, 1:500	HIER (pH 6)	Dako EnVision FLEX	Abcam (ab217676)
EMMPRIN	Mouse, monoclonal, clone 8D6, 1:100	HIER (pH 6)	Dako EnVision FLEX	Abcam (ab194401)

**Table 2 cancers-15-04924-t002:** Descriptive patients’ clinical characteristics. Yes is indicated by a +, while no is indicated by a -.

N	Sex	Age	OSCC Localization	pT	pN	pM	AJCC Stage	G	Dead	OS (Years)	Recurrence	DFS [Months]
1	F	76	Cheek	1	0	0	I	2	+	3	-	24
2	M	55	Tongue	1	0	0	I	1	-	4	+	12
3	M	57	Gum	2	2	0	IV	2	+	2	-	22
4	M	89	Tongue	2	2	0	IV	2	+	2	+	0
5	M	78	Gum	2	2	0	IV	2	+	1	+	8
6	F	78	Gum	2	0	0	II	2	-	4	+	6
7	M	44	Floor of mouth	1	0	0	I	2	-	3	+	4
8	M	56	Palate	2	0	0	II	2	-	4	-	52
9	M	82	Palate	4	1	0	IV	2	+	1	+	6
10	F	65	Floor of mouth	2	0	0	II	2	-	4	-	34
11	M	58	Floor of mouth	1	0	0	I	2	-	5	+	26
12	F	84	Gum	2	0	0	II	2	+	1	+	4
13	M	63	Gum	3	0	0	III	2	+	2	+	5
14	M	45	Floor of mouth	4	2	0	IV	2	+	1	-	4
15	F	62	Inside of lips	3	2	0	IV	2	+	1	+	8
16	M	79	Floor of mouth	4	1	0	IV	2	+	1	-	9
17	M	74	Tongue	1	1	0	III	2	-	5	-	39
18	F	70	Gum	4	1	0	IV	2	-	5	+	2
19	F	49	Palate	4	2	0	IV	2	+	2	-	19
20	M	66	Floor of mouth	2	0	0	II	3	-	4	+	13
21	F	82	Gum	4	0	0	IV	2	-	3	-	35
22	F	63	Gum	4	2	0	IV	2	+	2	+	8
23	M	70	Floor of mouth	2	0	0	II	1	-	5	-	45
24	F	79	Floor of mouth	2	0	0	II	2	+	4	+	48

**Table 3 cancers-15-04924-t003:** H-score values derived from the semiquantitative immunohistochemical evaluation of CX43 expression in all ROIs. H-score values range from 0 to 300, where 0 means no cells are positive and 300 means all cells are strongly positive. OM = oral mucosa; OSCC = center of primary OSCC; IF = invasive front; LNM = lymph node metastasis.

N	OM	OSCC	IF	LNM
1	66	162	128	
2	170	182	132	
3	3	165	166	72
4	1	30	10	175
5	1	133	100	82
6	40	233	106	
7	38	115	89	
8	6	142	44	
9	3	191	115	
10	35	61	43	
11	1	69	57	
12	0	175	88	
13	0	0	0	
14	3	43	0	18
15	16	175	60	166
16	0	0	0	80
17	38	231	18	118
18	0	32	11	152
19	0	31	0	182
20	8	0	8	
21	0	188	76	
22	137	274	194	193
23	0	2	4	
24	25	20	30	

**Table 4 cancers-15-04924-t004:** H-score values derived from semiquantitative immunohistochemical evaluation of EMMPRIN expression in all ROIs. The H-score value spectrum ranges from 0 to 300, where 0 means no cells are positive and 300 means all cells are strongly positive. OM = oral mucosa; OSCC = center of primary OSCC; IF = invasive front; LNM = lymph node metastasis.

N	OM	OSCC	IF	LNM
1	19	195	237	
2	43	36	73	
3	47	119	213	206
4	9	40	75	0
5	0	141	256	151
6	150	213	274	
7	73	81	146	
8	16	180	192	
9	6	160	225	
10	8	45	95	
11	43	102	150	
12	41	54	107	
13	0	2	10	
14	11	33	36	70
15	0	0	0	30
16	1	193	78	219
17	72	110	229	0
18	46	27	41	129
19	28	57	59	110
20	0	52	242	
21	0	157	224	
22	0	267	291	0
23	28	86	153	
24	0	77	198	

**Table 5 cancers-15-04924-t005:** H-score values derived from semiquantitative immunohistochemical evaluation of E-cadherin expression in all ROIs. H-score values range from 0 to 300, where 0 means no cells are positive and 300 means all cells are strongly positive. OM = oral mucosa; OSCC = center of primary OSCC; IF = invasive front; LNM = lymph node metastasis.

N	OM	OSCC	IF	LNM
1	174	171	81	
2	102	107	17	
3	112	79	23	128
4	62	104	122	201
5	92	134	78	97
6	61	31	7	
7	244	217	113	
8	151	183	94	
9	109	118	40	
10	153	121	31	
11	117	85	82	
12	166	179	66	
13	49	65	66	
14	123	235	180	145
15	68	169	96	100
16	134	87	20	250
17	67	73	1	197
18	42	44	28	147
19	178	70	60	192
20	92	73	10	
21	107	136	116	
22	160	109	83	286
23	17	110	0	
24	18	8	0	

**Table 6 cancers-15-04924-t006:** H-score values derived from semiquantitative immunohistochemical evaluation of vimentin expression in all ROIs. H-score values range from 0 to 300, where 0 means no cells are positive and 300 means all cells are strongly positive. OM = oral mucosa; OSCC = center of primary OSCC; IF = invasive front; LNM = lymph node metastasis.

N	OM	OSCC	IF	LNM
1	27	27	94	
2	27	29	74	
3	45	134	189	125
4	30	49	76	84
5	25	57	96	113
6	12	38	114	
7	21	41	67	
8	22	39	73	
9	15	96	154	
10	17	38	102	
11	20	41	48	
12	25	79	108	
13	21	43	65	
14	9	12	13	50
15	1	16	193	53
16	41	60	104	33
17	80	54	76	112
18	47	54	86	65
19	69	47	65	37
20	25	29	94	
21	22	19	34	
22	19	14	28	10
23	100	69	174	
24	44	66	193	

**Table 7 cancers-15-04924-t007:** Significant effects in the first univariable screening analysis for DFS. Δ indicates change in expression between ROIs; OM = oral mucosa; IF = invasive front. Hazard ratios (HRs) are presented for each factor, along with the 95% confidence interval (CI) and associated *p*-value.

Variable	Level	N	HR	95% CI	*p*-Value
Age	≤45	2			
	>45	22	0.11	[0.02; 0.69]	0.018
Adjuvant therapy	No	14			
	Yes	10	3.9	[1.40; 11.0]	0.01
ΔE-cad (OM-IF)	≤9	20			
	>9	4	6	[1.70; 22.0]	0.006
Cx43 (IF)	≤75.7	15			
	>75.7	9	2.7	[1.0; 7.20]	0.045
EMMPRIN (IF)	≤146	11			
	>146	13	0.3	[0.11; 0.79]	0.014
ΔEMMPRIN (OM-IF)	≤77	10			
	>77	14	0.15	[0.05; 0.47]	0.001

**Table 8 cancers-15-04924-t008:** Screening of protein expression levels as potential risk factors for DFS, adjusted for AJCC stage and age. Δ indicates change in expression between ROIs; OM = oral mucosa; OSCC = center of primary OSCC; IF = invasive front. Hazard ratios (HRs) are presented for each factor, along with the 95% confidence interval (CI) and associated *p*-value.

Variable	Level	N	HR	95% CI	*p*-Value
EMMPRIN (OSCC)	≤40	6			
	>40	18	0.16	[0.04; 0.56]	0.004
EMMPRIN (IF)	≤146	11			
	>146	13	0.27	[0.09; 0.77]	0.014
ΔEMMPRIN (OM-IF)	≤77	10			
	>77	14	0.13	[0.04; 0.46]	0.001
ΔE-cad (OM-IF)	≤9	20			
	>9	4	5.2	[1.24; 21.7]	0.024
Cx43 (IF)	≤75.7	15			
	>75.7	9	2.7	[1.00; 7.10]	0.05

**Table 9 cancers-15-04924-t009:** Multivariable model for DFS. Δ indicates expression change between ROIs; OM = oral mucosa; IF = invasive front. Hazard ratios (HRs) are presented for each factor, along with the 95% confidence interval (CI) and associated *p*-value.

Variable	N	HR	95% CI	*p*-Value
ΔEMMPRIN (OM-IF)	24	0.99	[0.99; 1.00]	0.019
ΔE-cad (OM-IF)	24	1.01	[1.00; 1.00]	0.254
Cx43 (IF)	24	1.01	[1.00; 1.00]	0.107

**Table 10 cancers-15-04924-t010:** Significant effects of univariable screening for potential risk factors for OS. Δ indicates change in expression between ROIs; OM = oral mucosa; OSCC = center of primary OSCC; IF = invasive front; LNM = lymph node metastasis. Hazard ratios (HRs) are presented for each factor, along with the 95% confidence interval (CI) and associated *p*-value.

Variable	Level	N	HR	95% CI	*p*-Value
pT	pT ≤ 2	15			
	pT > 2	9	3.4	[1.10; 10.0]	0.032
pN	N−	13			
	N+	11	5.2	[1.50; 17.0]	0.008
Adjuvant therapy	No	14			
	Yes	10	6.8	[1.90; 24.0]	0.003
AJCC stage	≤2	11			
	>2	13	5.3	[1.40; 20.0]	0.014
ΔE-cad (IF-LNM)	≤−78.8	7			
	>−78.8	3	13	[1.20; 128.0]	0.032
ΔE-cad (OM-IF)	≤9	20			
	>9	4	4.8	[1.30; 17.0]	0.016
Vim (OM)	≤9	2			
	>9	22	0.1	[0.02; 0.59]	0.011
Vim (OSCC)	≤54	17			
	>54	7	3.4	[1.10; 10.0]	0.028
Δvim (OM-OSCC)	≤−32.4	4			
	>−32.4	20	0.14	[0.04; 0.52]	0.003
Δvim (OM-IF)	≤102	20			
	>102	4	3.3	[1.0; 11.0]	0.047
Cx43 (IF)	≤0	4			
	>0	20	0.21	[0.06; 0.74]	0.016
ΔCx43 (OM-IF)	≤75.7	20			
	>75.7	4	7.1	[1.90; 26.0]	0.003
EMMPRIN (OM)	≤13.5	12			
	>13.5	12	0.29	[0.09; 0.94]	0.039
EMMPRIN (IF)	≤36	3			
	>36	21	0.19	[0.05; 0.75]	0.018
ΔEMMPRIN (OSCC-IF)	≤−12	19			
	>−12	5	6.4	[1.80; 22.0]	0.003

**Table 11 cancers-15-04924-t011:** Screening of protein expression levels as potential risk factors for DFS, controlling for age and AJCC stage. Δ indicates change in expression between ROIs; OM = oral mucosa; OSCC = center of primary OSCC; IF = invasive front. Hazard ratios (HRs) are presented for each factor, along with the 95% confidence interval (CI) and associated *p*-value.

Variable	Level	N	HR	95% CI	*p*-Value
ΔEMMPRIN (OSCC-IF)	≤−12	19			
	>−12	5	6.9	[1.40; 34.5]	0.018
Δvim (OM-OSCC)	≤−32.4	4			
ΔCx43 (OM-IF)	>−32.4	20	0.21	[0.06; 0.81]	0.024
	≤75.7	20			
	>75.7	4	4.7	[1.23; 18.0]	0.024

**Table 12 cancers-15-04924-t012:** Multivariable model for OS. Δ indicates expression change between ROIs; OM = oral mucosa; IF = invasive front. Hazard ratios (HRs) are presented for each factor, along with the 95% confidence interval (CI) and associated *p*-value.

Variable	N	HR	95% CI	*p*-Value
ΔEMMPRIN (OSCC-IF)	24	1.02	[1.00; 1.00]	0.023
Δvim (OM-OSCC)	24	0.98	[0.95; 1.00]	0.240
ΔCx43 (OM-IF)	24	1.01	[0.99; 1.00]	0.603

## Data Availability

All data can be obtained from the corresponding author.

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
