# Peer review of "Combined Biomarker System Predicts Prognosis in Patients with Metastatic Oral Squamous Cell Carcinoma"

_cancers, 2023, doi:10.3390/cancers15204924_

Round 1

Reviewer 1 Report

1. This study is a great innovation in independently evaluating protein expression levels in different types of specimens and different parts of tumor samples. It helps to comprehensively understand the invasion front of tumors and the protein expression profile changes of metastatic lymph nodes during tumor progression and metastasis, and evaluate the prognosis of tumors based on this. However, there are still some issues that need improvement.

2. The author should provide raw data to validate the results, such as semi quantitative values of immunohistochemistry (all ROIs of each sample), and time points of recurrence or death for each patient.

3. Please provide the product number and brand of the antibodies in Table 1.

4. In lines 282-322, the univariable/multivariable COX analysis for the dependent variables DFS and OS should be presented in the form of a table or a combined forest map. Each variable should have HR values and confidence intervals noted, and a simple textual description may seem very difficult for readers to understand and lack credibility.

5. For the Combined Biomarker System, the author can present the specific situation of prognosis prediction in the form of a comprehensive score that conforms to the forest map at the end. Furthermore, the necessary independent validation dataset can be presented to further validate the reliability of this Combined Biomarker System.

Author Response

Reviewer 1

2.) The author should provide raw data to validate the results, such as semi quantitative values of immunohistochemistry (all ROIs of each sample), and time points of recurrence or death for each patient.

Authors respond: We would like to thank reviewer 1 for the critical evaluation of the manuscript and the useful suggestions to improve the quality of the paper. We have followed the recommendation and included the raw data (H-score values) of the semiquantitative immunohistochemical evaluation in the results sections of the corresponding marker proteins. In addition, we decided to completely redesign Table 2 to include the detailed clinical and pathologic parameters of each patient. For the sake of clarity, we have reported the time to recurrence (disease-free survival; DFS) and overall survival (OS). The changes can be found at the appropriate place in the manuscript. In addition, we have made the complete underlying data set available online. This can be accessed via the following link: https://doi.org/10.25625/FNR4EX. A corresponding note has been added to the manuscript.

3.) Please provide the product number and brand of the antibodies in Table 1.

Authors respond: We have revised Table 1 and included the corresponding product data in the manuscript.

4.) In lines 282-322, the univariable/multivariable COX analysis for the dependent variables DFS and OS should be presented in the form of a table or a combined forest map. Each variable should have HR values and confidence intervals noted, and a simple textual description may seem very difficult for readers to understand and lack credibility.

Authors respond: At the suggestion of the reviewer, we have decided to present all significant parameters from both the initial univariable screening analyses and the multivariable analysis for DFS and OS in the form of tables with HR and 95% CI at the appropriate places in the manuscript. In contrast to a forest plot, additional information such as the corresponding cut-off values can be included for easier interpretation. Due to the large number of factors tested and for clarity, we have omitted the non-significant parameters when presenting the univariable analyses.

5.) For the Combined Biomarker System, the author can present the specific situation of prognosis prediction in the form of a comprehensive score that conforms to the forest map at the end. Furthermore, the necessary independent validation dataset can be presented to further validate the reliability of this Combined Biomarker System.

Authors respond: Thank you for this useful comment. All protein variables with a P value <0.1 in the initial univariable screen were reassessed for association with DFS and OS adjusted for AJCC stage and age. The number of patients and events limited the complexity of the models that could be fitted to the data. The variables adjuvant therapy, pT, pN, and AJCC stage were highly correlated, so AJCC stage was chosen as representative of this group of variables. We chose to adjust for AJCC stage because it integrates all individual TNM factors. Factors that were independent of clinical parameters in this analysis were included in the final multivariable regression model. To examine the model predictions of the combined marker system, the multivariable Cox regression model was fitted per prognostic endpoint (DFS and OS) according to the following equations:

Score Formula DFS:

0.73805 - 0.00759 * ΔEMMPRIN (OM-IF) + 0.00576 * ΔE-Cadherin (OM-IF) + 0.00783 * Cx43 (IF)

Score Formula OS:

0.21709 + 0.01490 * ΔEMMPRIN (OCSS-IF) - 0.01966 * ΔVim (OM-OSCC) + 0.00531 * ΔCx43 (OM-IF)

To visualize the results, model predictions were binarized based on both median and maximum selected rank statistics, and Kaplan-Meier curves were plotted in the resulting subgroups and compared using log-rank tests. We believe this approach is the best and clearest way to demonstrate the combined prognostic effect with the data available to date. We have included the appropriate references in the manuscript. We agree that validation of the results with an independent data set is necessary. However, to the best of our knowledge, there is no comparable study in the literature that has compared the protein expression profiles of the various marker proteins in the different regions of interest in the metastatic process of OSCC in such an individualized manner. Therefore, validation using data from public databases is not feasible or meaningful. We plan to validate the results of this initial screening study in subsequent functional studies. However, we hope to have drawn attention to the altered expression profiles during the metastatic process of OSCC in order to further elucidate the previously unclear background factors in the future.

Reviewer 2 Report

This manuscript explores the expression profiles of key proteins involved in the metastatic progression of oral squamous cell carcinoma (OSCC). The study involved 24 OSCC patients, and the researchers assessed the expression of connexin 43 (Cx43), extracellular matrix metalloproteinase inducer (EMMPRIN), E-cadherin, and vimentin in different regions of interest (ROIs), including healthy oral mucosa, the center of primary OSCC, the invasive front, and local lymph node metastasis (LNM). The study revealed distinct protein expression patterns across these ROIs and the researchers performed prognostic analysis. Additionally, it proposed a combined biomarker system involving these proteins for predicting patient outcomes. In conclusion, this research underscores the importance of EMMPRIN expression in OSCC metastasis and its potential as a therapeutic target. The combined biomarker system comprising Cx43, EMMPRIN, E-cadherin, and vimentin demonstrates promise for predicting DFS and OS in OSCC patients, offering potential clinical utility in guiding treatment decisions.

However, certain key questions remain:

1. Validation of the immunohistochemistry (IHC) results through alternative techniques such as Western blotting or qRT-PCR is advisable to provide more precise quantification of protein expression levels.

2. Exploration of the study's findings in a larger cohort of OSCC patients, possibly from publicly available databases, for clinical validation would enhance the study's robustness and generalizability.

Author Response

Reviewer 2

1.) Validation of the immunohistochemistry (IHC) results through alternative techniques such as Western blotting or qRT-PCR is advisable to provide more precise quantification of protein expression levels.

Authors respond: We would like to thank reviewer 2 for the critical review of the manuscript and useful suggestions. The aim of this initial screening analysis was to obtain information on the changes in the protein expression profiles of the four marker proteins associated with the EMT process (Cx43, EMMPRIN, E-cadherin, and vimentin) during the metastatic pathway of OSCCs and to correlate them with patient prognosis. No meaningful qRT-PCR or Western blot analysis can be performed on the preserved paraffin-embedded tissue material from patients with different proportions of tumor and associated stroma. However, the results of this study provide important information to evaluate the biological background in further functional investigations, and plans to perform these studies are in preparation. We have revised the manuscript on the advice of the reviewers and added the raw data of the respective simi-quantitative immunohistochemical evaluation to the individual marker proteins. In addition, we have redesigned Table 2 to reflect the individual clinicopathological factors of the patients in detail. In the prognostic analysis section, the results of univariable and multivariable analyses are reported in the form of tables with HR and 95% CI. In addition, we have made the complete underlying data set available online. This can be accessed via the following link: https://doi.org/10.25625/FNR4EX. All changes can be found in the revised manuscript.

2.) Exploration of the study’s findings in a larger cohort of OSCC patients, possibly from publicly available databases, for clinical validation would enhance the study’s robustness and generalizability.

Authors respond: To the best of our knowledge, there is no comparable study in the literature to date that has examined the protein expression profiles of the various marker proteins in the different regions of interest in the OSCC metastatic process in such an individualized manner. Therefore, comparison with other data from public databases is not feasible or useful. Before designing the study, we performed a statistical power analysis to calculate the minimum number of patients required to reach a significant conclusion. The results of this initial screening study will be evaluated in subsequent functional studies. However, we hope to have drawn attention to the altered expression profiles during the metastatic process in OSCC in order to elucidate the previously unclear background factors in detail in the future.

Reviewer 3 Report

This is an interesting article discussing to role of a combined protein markers system for the prognosis and prediction of metastatic OCSCC. This was an important point for the prognosis and prediction of metastatic OCSCC. However, there were several limitations as follows.

1.      The enrolled patients were only 24. This was a small population to establish a prediction model

2.      In Table 2, there were several important pathologic features not presented.

3.      As the prognostic role, the markers were not adjusted by TNM staging, major pathologic features (positive margin, extracapsular extension), and other pathologic features. Please show the table as a univariant and multivariate Cox regression analysis, both for DFS and OS

Extensive editing of English language required

Author Response

Reviewer 3

1.) The enrolled patients were only 24. This was a small population to establish a prediction model.

Authors respond: The authors would like to thank reviewer 3 for reviewing the manuscript and providing useful comments. Prior to the initiation of this study, a statistical power analysis was performed to determine the minimum number of patients required to reach a significant conclusion. Because histologic staining for all four marker proteins (Cx43, EMMPRIN, E-cadherin, and vimentin) and additional HE staining for morphologic assessment of the primary OSCC and adjacent oral mucosa were performed in each of the 24 patients, and all five stains were also performed on tissue samples from local cervical lymph node metastases in half of the patients, the processing volume was high.  The study should be considered as an initial screening to evaluate protein expression changes through the OSCC metastatic pathway and the combined marker system. Important conclusions for future functional studies were drawn. To our knowledge, such an individualized study has not been published in the literature. The impact of the EMMPRIN expression profile provides relevant information that may be of interest for a targeted therapeutic approach in the future. We have also had the manuscript linguistically revised by a professional editing service (certificate provided).

2.) In Table 2, there were several important pathologic features not presented.

Authors respond: We would like to thank you for this useful suggestion. Based on the reviewer's comment, we have completely redesigned Table 2, which now lists the main clinical and pathological parameters for each patient. The changes can be found at the appropriate place in the revised manuscript.

3.) As the prognostic role, the markers were not adjusted by TNM staging, major pathologic features (positive margin, extracapsular extension), and other pathologic features. Please show the table as a univariant and multivariate Cox regression analysis, both for DFS and OS.

Authors respond: We have included all significant parameters of the univariable and multivariable analyses in tabular form in the manuscript, as recommended. Due to the large number of factors examined and for the sake of clarity, we have presented only the significant factors. In addition, we have made the complete underlying data set available online. This can be accessed via the following link: https://doi.org/10.25625/FNR4EX. An appropriate note has been added to the manuscript. It is not entirely true that the factors were not adjusted for TNM classification. All protein variables with a P value < 0.1 in the initial univariable screening were re-screened for association with DFS and OS adjusted for AJCC stage and age. The number of patients and events limited the complexity of the models that can be fitted to the data. The variables adjuvant therapy, pT, pN, and AJCC stage were highly correlated, so AJCC stage was chosen as a representative of this group of variables. We decided to adjust for AJCC stage because it integrates all the individual TNM factors. Only those variables that showed an independent prognostic impact in this analysis were included in the final multivariable analysis. R0 margin status was achieved in all patients based on intraoperative frozen section diagnosis. Examination of each individual lymph node status would not be useful due to the small number of different N stages and the large number of statistical tests. For reliable statistical analysis, patients were divided into pN+ and pN- groups.

Round 2

Reviewer 3 Report

Thanks for the authors's revision. I think the manuscript was well to be accepted